# An Extracellular/Membrane-Bound S100P Pool Regulates Motility and Invasion of Human Extravillous Trophoblast Lines and Primary Cells

**DOI:** 10.3390/biom13081231

**Published:** 2023-08-09

**Authors:** Tara Lancaster, Maral E. A. Tabrizi, Mariaelena Repici, Janesh Gupta, Stephane R. Gross

**Affiliations:** 1College of Health and Life Sciences, Aston University, Birmingham B4 7ET, UK; taralancaster94@gmail.com (T.L.); m.ebrahimzadehasltabrizi2@aston.ac.uk (M.E.A.T.); m.repici@aston.ac.uk (M.R.); 2Institute of Metabolism and Systems Research, The University of Birmingham, Birmingham B15 2TT, UK; j.k.gupta@bham.ac.uk; 3Fetal Medicine Centre, Birmingham Women’s NHS Foundation Trust, Birmingham B15 2TT, UK

**Keywords:** S100P, membrane, trophoblasts, migration, invasion, cytoskeleton, placenta

## Abstract

Whilst S100P has been shown to be a marker for carcinogenesis, we have shown, in non-physio-pathological states, that its expression promotes trophoblast motility and invasion but the mechanisms explaining these cellular processes are unknown. Here we identify the presence of S100P in the plasma membrane/cell surface of all trophoblast cells tested, whether lines, primary extravillous (EVT) cells, or section tissue samples using either biochemical purification of plasma membrane material, cell surface protein isolation through biotinylation, or microscopy analysis. Using extracellular loss of function studies, through addition of a specific S100P antibody, our work shows that inhibiting the cell surface/membrane-bound or extracellular S100P pools significantly reduces, but importantly only in part, both cell motility and cellular invasion in different trophoblastic cell lines, as well as primary EVTs. Interestingly, this loss in cellular motility/invasion did not result in changes to the overall actin organisation and focal adhesion complexes. These findings shed new light on at least two newly characterized pathways by which S100P promotes trophoblast cellular motility and invasion. One where cellular S100P levels involve the remodelling of focal adhesions whilst another, an extracellular pathway, appears to be focal adhesion independent. Both pathways could lead to the identification of novel targets that may explain why significant numbers of confirmed human pregnancies suffer complications through poor placental implantation.

## 1. Introduction

During the development of the human placenta, a process also known as placentation, a subset of trophoblast cells proliferates and differentiates into an extravillous trophoblast (EVT) population. Their primary function is to invade the maternal decidua and myometrium and promote the organization of the maternal spiral arteries [1], a step which is key for embryogenesis. As a result, poor invasion by EVT is thought to lead to pregnancy pathologies and complications such as preeclampsia, foetal growth restriction, and miscarriage in the most extremes of cases [2]. Due to the importance of EVTs’ invasion in the early stages of gestation, active searches have aimed to establish the role of invasiveness-promoting factors and a myriad of active molecules acting as growth factors [3,4], metalloproteinases [5], or receptor/adhesion molecules [6], to cite just a few, have been identified. Interestingly, migration and invasion characteristics of trophoblast cells have been likened to those seen in cancer cells. Consequently numerous proteins, whose functions have been highlighted as markers for carcinogenesis, have been reported to be important regulators of trophoblast migration/invasion (recently reviewed [7]), of which ezrin [8] and S100P [9] are two of the newest additions.

S100P expression has been shown to be correlated with malignancy progression of tumour cells originating from numerous tissue sources, such as lung, pancreas, and breast [10,11,12,13,14,15]. The specific expression of S100P has been shown to be essential for acquiring changes in cellular motility and invasion [16,17,18,19]. How S100P promotes cellular motility and invasion in different cells has been put forward in the context of cancer migration/invasion [20]. Intracellular partners such as myosin IIA [16], ezrin [17], IQGAP1 [21], and microtubules [22] have been shown to interact with S100P and to affect their overall properties, possibly highlighting processes which lead to migration/invasion changes. We have recently shown that S100P also activates tissue plasminogen activator in breast cancer cells and that the *C*-terminal lysine of S100P is essential for enhancing cell migration and invasiveness [23,24]. However, it is unclear how S100P regulates such processes in non-physio-pathological conditions such as in trophoblast cells during placentation.

Expression of S100P in trophoblast cells both in vivo and in cell lines has been reported [9,25,26,27]. This protein was in fact first discovered in the placenta [28,29] where it is found at a concentration 90–200-fold higher than in any other organs [30]. We have recently shown that both gain and loss of S100P functions resulted in increased or decreased abilities of trophoblasts cells motility and invasion, respectively, whereby focal adhesion dynamics are affected [9], but the clear mechanisms for this remain to be fully characterised.

In this work we aimed to gain further understanding and shed new light onto some of the molecular pathways that are being regulated by S100P expressions in trophoblast cells. We show here for the first time, using both biochemical and microscopy analysis, that some pools of S100P protein can be found localised at the extracellular surface of the plasma membrane in different trophoblast cells including lines and primary cells. Further analysis demonstrates that inhibiting the extracellular cell membrane associated levels of S100P by addition of specific antibody leads to dramatic reduction in trophoblast motility and invasion without significant changes in cytoskeletal remodelling/focal adhesion. This suggests that this extracellular membrane associated pool of S100P promotes motility through a different mechanism that has previously been reported [9]. Our data therefore demonstrates a new function for S100P which links its well-characterised motility and invasion, which enhances properties of the physiological placentation process.

## 2. Materials and Methods

### 2.1. Human Placental Tissues

Written informed consent was obtained from all women recruited into the study. Samples of placenta tissues for EVTs isolation were obtained immediately after elective termination of pregnancy from first trimester of gestation (8–12 weeks). Placental samples were collected with approval of Health Research Authority—West Midlands, Edgbaston Research Ethics Committee (NHS REC 15/WM/0284 and AHRIC REF 1245-SG, Nottingham, UK). Tissue used for immunohistochemical staining (*n* = 3) was collected with approval by the National Research Ethics Service (NRES) Committee North West—Haydock (study approval number 13/NE/2005, Manchester, UK), at 6 weeks gestation. Samples were fixed in 4% (*w*/*v*) paraformaldehyde and embedded in paraffin wax prior to further processing. All samples and tissues were collected in accordance with relevant guidelines and regulations.

### 2.2. Cell Lines and Culture

Two human placental derived choriocarcinoma trophoblastic cell lines Jeg-3 and Bewo were used as they have been shown to endogenously express different levels of S100P [9], and were grown in MEM medium (Sigma, Hertfordshire, UK) or F-12 medium (Sigma, Hertfordshire, UK), respectively, supplemented with 10% (*v*/*v*) Fetal Bovine Serum (FBS; Sigma, Hertfordshire, UK), 100 units and 0.1 mg/mL penicillin/streptomycin (Sigma, Hertfordshire, UK), respectively, and 2 mM L-glutamine (Sigma, Hertfordshire, UK) as previously described [9].

The HTR8/SVneo first trimester extravillous trophoblast cells (kind gift of Prof. Graham Charles (Queen’s University, Kingston, ON, Canada) engineered to express high levels of S100P proteins (two independent clones referred to as Clone 5 and Clone 7) or the S100P-negative control counterpart (clone 3) have been characterised and reported previously [9]. Cells were grown in RPMI (Sigma, Hertfordshire, UK) supplemented with 5% (*v*/*v*) FBS, 100 units and 0.1mg/mL penicillin/streptomycin, respectively, and 2 mM L-glutamine, and 50 μg/mL hygromycin B.

For the inhibition of S100P, either polyclonal goat anti S100P antibody (R&D Systems, Manchester, UK) or S100P siRNA (FlexiTube S100P siRNA 4 (QIAGEN, Manchester, UK)) were used. S100P antibody (dilution of 1:1000) was added prior to seeding whilst delivery of siRNA (5 nM) was carried out for 48 h prior to seeding. Controls were also used for the corresponding treatments as previously described [9].

### 2.3. EVT Purification

The protocol for isolating EVTs has been previously described [8] and adapted from [31]. After washing first trimester placental samples in Ham’s F12 (Sigma, Hertfordshire, UK), chorionic villi were physically separated and digested in 0.25% (*w*/*v*) trypsin solution (Sigma, Hertfordshire, UK). The placental tissue was then filtered and diluted in 25% (*v*/*v*) FBS in Ham’s F12 medium (Sigma, Hertfordshire, UK) prior to centrifugation for 5 min at 450× *g*. Pellets were resuspended and collected prior to being layered over a Pancoll solution (Pan Biotech, Aidenbach, Germany. Density 1.077 g/mL). Samples were spun at 750× *g* for 20 min to allow for concentration and isolation of the EVT band which was then aspirated and collected into clean tubes prior to a final centrifugation at 500× *g*. The resulting EVTs were resuspended in trophoblast complete medium (TCM; Ham’s F12 without Phenol red, 20% (*v*/*v*) FBS, 100 units and 0.1 mg/mL penicillin/streptomycin, 2 mM L-Glutamine). These cells were then seeded onto 35 mm tissue culture dishes and left to settle overnight before changing the medium to fresh TCM.

### 2.4. Plasma Membrane Isolation

Cells were grown to confluency in four 10 cm dishes, washed twice with PBS then scraped into PBS and centrifuged at 1350× *g* for 5 min. Following removal of the supernatant, the pellet was resuspended in homogenisation buffer (250 mM sucrose, 50 mM Tris, 0.25 mM CaCl_2_ pH 7.4), and centrifuged twice at 500× *g* for 5 min. The cell pellet was resuspended in homogenisation buffer and passed through the cell disruption bomb chamber (Parr instruments, Moline, IL, USA) at 4 °C. Cells were equilibrated in the cell disruption bomb chamber at 800–1000 PSI for 20 min. Following release from the bomb, the suspension was centrifuged at 750× *g* for 10 min to remove remaining whole cells and nuclei. The supernatant was collected and layered over 35% (*w*/*w*) sucrose/50 mM Tris and centrifuged at 100,000× *g* for 1 h. The interface was collected and topped up with 25 mM sucrose/50 mM Tris pH 7.4 before spinning at 100,000× *g* for 30 min. The pellet containing plasma membranes was resuspended in 250 mM sucrose/50 mM Tris pH 7.4. Samples were stored at −20 °C prior to analysis via Western blot as previously described [8,9]. Original, uncropped, and unadjusted images are available as Appendix A. Quantification was obtained following densitometry analysis of at least 3 independent experiments and presented to correspond to the percentage of signal obtained from the membrane compared to the proportion seen in the total cell extract (for tubulin or S100P depending on the experiments carried out).

### 2.5. Biotinylation

Cell surface proteins were biotinylated and extracted using the Cell Surface Isolation Kit (Pierce, Liverpool, UK) according to the manufacturer’s instructions. Briefly, cells were grown to 90% confluency, washed twice with ice-cold PBS prior to addition of 250 µg/mL Sulfo-NHS-SS-Biotin and incubation on ice, for 30 min. Quenching solution was added and cells were scraped and centrifuged at 500× *g* for 3 min. The cell pellet was resuspended in TBS and again centrifuged at 500× *g* for 3 min prior to addition of lysis buffer. Cells were resuspended in TBS and sonicated on ice and left to incubate on ice for 30 min. The lysate was clarified by centrifugation at 10,000× *g* for 2 min at 4 °C prior to loading on NeutrAvidin agarose and incubation for 60 min at room temperature. Following multiple washes, bound proteins were eluted from the columns using sample buffer (62.5 mM Tris HCl pH 6.8, 1% SDS, 10% glycerol, 50 mM DTT) on a heat block for 5 min at 95 °C prior to collection of samples by centrifugation for 2 min at 1000× *g*. Samples were stored at −20 °C prior to analysis via Western blot as previously described [8,9]. Original, uncropped, and unadjusted images are available as Appendix A.

### 2.6. Immunofluorescent Staining

Immunofluorescence was carried out as previously described [8,9,32]. Briefly all trophoblast cell lines (Jeg-3, Bewo and HTR8/SVneo) as well as primary human first trimester EVTs were plated at a concentration of 20,000 cells/well onto fibronectin-coated (2.5 μg/cm^2^) glass coverslips in a 24-well plate for 24 h prior to washing with cytoskeleton buffer (CB). Cells were then fixed with 3.7% (*w*/*v*) paraformaldehyde in CB at 37 °C for 10 min followed by permeabilisation with 0.1% (*v*/*v*) Triton X-100 in CB for 10 min and blocking with blocking solution (10% (*v*/*v*) goat serum in CB) for 30 min. Incubations with primary antibodies against S100P, Integrin b1 or paxillin (Appendix A) were carried out for 45 min at room temperature after dilution in blocking solution (1% (*v*/*v*) goat serum in CB). Following three washes with 1% (*v*/*v*) goat serum in CB, cells were incubated with the appropriate secondary anti-rabbit or anti- mouse antibodies labelled with FITC or TRITC (Appendix A), respectively, in blocking solution for 45 min at room temperature. For actin staining, rhodamine phalloidin (Invitrogen, Paisley, UK) was also added with secondary antibodies at a concentration of 0.6 μM.

For membrane co-localization, the markers Nile Red staining or Integrin β1 were used. For the former, cells were incubated with S100P antibody (BD Biosciences (Appendix A) for 1 h at 37 °C, followed by another brief incubation with Nile red (300 nM) for 5 min prior to fixation. For the latter, cells were first fixed prior to incubation with S100P and Integrin β1 antibodies. For all conditions, the samples were incubated with the appropriate secondary anti-rabbit or anti- mouse antibodies labelled with FITC or TRITC (Appendix A), respectively, in blocking solution for 45 min at room temperature. Samples were then mounted in Vectashield mounting medium (Vector Labs, Peterborough, UK), and viewed using either an Epifluorescence Leica DMI400B microscope equipped with a 63× oil objective or a confocal Leica SP8 Falcon Film system equipped with an HC PL APO 40 × 0.95 DRY objective.

Images were analysed for the presence of S100P at the plasma membrane using ImageJ plot profile function https://imagej.nih.gov/nih-image/more-docs/Tutorial/Profile.html (accessed on 15 March 2019) or the Leica Application Suite X. Images after staining for S100P and Integrin β1 or Nile red were taken and regions of interest (ROI) were drawn across each cell, with the membrane being placed at the midpoint of the region. 

Quantification of focal adhesions following addition of S100P antibodies/S100P siRNA or in untreated samples was carried out by counting the number of focal adhesions per cell in randomly selected cells. Results are presented as means ± SD of 3 independent experiments. An average of 50 cells were counted per experimental condition.

### 2.7. Immunohistochemistry

Serial sections of human placental tissue were stained for S100P and human leukocyte antigen-G (HLA-G) as previously described [8,9]. Briefly, sample tissue sections were deparaffinised, and heat-induced antigen-retrieval was carried out for 10 min using citrate buffer (pH 6.0) by heating with a microwave at 800 Watts. Blocking was achieved after incubating sections with BLOXALL (Vector Laboratories, Peterborough, UK) for 10 min and then with 2.5% (*w*/*v*) goat serum for 30 min. Staining was carried out using monoclonal anti-S100P or anti-HLA-G antibodies (Abcam, Cambridge, UK) incubated at 4 °C overnight (Appendix A). Following washing in PBS, the appropriate conjugated second antibody (Appendix A) was used. Images were acquired using NDP software and the staining was analysed using a Nano Zoomer XR scanner (Hamamatsu, Shizuoka, Japan). Analysis for the presence of S100P and HLA-G at the plasma membrane was carried out using ImageJ plot profile function (https://imagej.nih.gov/ij/docs/examples/calibration/ (accessed on 20 June 2019)). Matched images after staining for S100P and HLA-G were taken, and the same region of interest (ROI) was drawn across each cell with the membrane being placed at the midpoint of the region. Signal density was measured across a grey scale to reflect concentration of the epitopes.

### 2.8. Motility/Invasion Assay

Motility and invasion abilities for all trophoblast cells were measured as previously described [8,9]. Briefly 10^5^ cells that had been serum-starved for 24 h were seeded in 0.5% (*v*/*v*) FBS-containing medium in wells with or without congealed matrigel (1:3 with serum free medium Sigma, Hertfordshire, UK) and 10% (*v*/*v*) FBS medium in the outer wells. Cells were allowed to migrate through the membranes for 24 h prior to being fixed and stained using May Grϋnwald/Giemsa staining (Sigma, Hertfordshire, UK). Stained cells on the upper surface of the membrane were removed and those on the lower side of the membrane were counted using a light microscope with a ×20 or ×40 objective lens, selecting 5 random fields. Data are presented as the mean ± SEM of 3 or 4 independent experiments relative to controls (percentage) from 4 replicate wells for each set of conditions. Images of representative fields of motility/invasion assays were taken with the EVOS XL Cell Imaging System at ×20 magnification.

### 2.9. Cell Counting and Viability Using Trypan Blue Exclusion

Jeg-3 and HTR8/SVneo clones were seeded at a density of 20,000 cells per well in 24 well plates and incubated for further timepoints between 24–48 h in the presence of S100P antibody. At each time point, cells were trypsinised (0.025% (*w*/*v*) trypsin in 2.5 mM EDTA) prior to re-equilibration in serum-containing medium, centrifugation, and resuspension in trypan blue. Total cell number was determined by trypan blue staining by counting in a haemocytometer. Data are presented as percentage of live cells (means ± SD of 3 independent experiments) relative to control.

### 2.10. Statistical Analysis

Results were analysed by one-way ANOVA followed by Tukey’s multiple comparison test using GraphPad Prism 7.03 (GraphPad Software, San Diego, CA, USA). Results are presented as the means ± SD or SEM. Means are considered significantly different from each other are indicated by * *p* < 0.05, *** *p* < 0.001 or **** *p* < 0.0001 or not significant (ns).

## 3. Results

### 3.1. S100P Is Localised to the Plasma Membrane of Human First Trimester EVTs In Vivo

Our previous work has shown that S100P is highly expressed in first trimester human placental samples and can be found in EVT in anchoring columns [9]. Here we aimed to analyse the cellular compartment(s) positive for S100P in cells using immunohistochemistry staining on paraffin-embedded sections from early gestational stages in relation to expression of the cell surface protein HLA-G [33], a known EVT marker. As expected, S100P signals were seen in EVTs in anchoring columns (Figure 1A). Analysis of the intensity of S100P staining (Figure 1B) demonstrated an increase in signal intensity near or around the plasma membrane region (Figure 1C), suggesting that in first trimester EVT, some S100P pools maybe localised in close proximity to this cellular compartment.

### 3.2. Pools of S100P Are Localised in the Region of the Plasma Membrane in Human Trophoblast Cell Lines and First Trimester EVTs Trophoblasts in Culture

We next wanted to determine whether the membrane subcellular localisation thought to have been seen in trophoblast cells during the first trimester of human placental development could also be found in cell lines expressing S100P as well as primary cells. This was achieved using immunofluorescence analysis with different markers (Figure 2). Specificity of staining for S100P was confirmed by the absence of signal when probing for S100P in the HTR8/SVneo clone 3 (non S100P expressing cells) whether with or without fixation (Figure 2A,B), since we have previously shown by Western blotting/mRNA that this clone has no detectable levels of S100P [9].

In S100P expressing cells, the majority of the signal was found either in the nuclear regions or the cytoplasmic compartments, but some S100P pools were found to be localised in/or near the regions of the plasma membrane of chorionic trophoblast-derived cell lines Jeg-3 and BeWo in addition to EVT-derived cell line HTR8/SVneo clone 5 and clone 7 cells (Arrowheads in Figure 2A).

Further experiments to show the presence of S100P in or near the plasma membranes were conducted when S100P antibody was added to the medium, along with the plasma membrane marker Nile red [34,35,36] prior to fixation (Figure 2B). In parallel, experiments to analyse the colocalisation of S100P staining with integrin β1, a well-established marker for plasma membrane in trophoblast cells [5,37,38] were also conducted in Jeg-3 cells as well as in primary EVT isolated from first trimester placenta (Figure 2C). Samples were analysed using selected membrane region of interest for signal profiles (Figure 2B lower panels and Figure 2D). In all cases, some colocalisation could be seen between S100P and either the marker Nile red or the integrin β1 receptor in regions close to the plasma membrane with fluorescence profiling supporting these observations (Figure 2B lower panels and Figure 2D).

Taken together these data suggest that a pool of S100P proteins can be found localised near to the plasma membrane domains of all trophoblast cells studied whether cell lines, primary cells, or in tissues.

### 3.3. S100P Is Found Associated at the Cell Surface of the Plasma Membrane in Trophoblast Cells

Next, we sought to confirm association of S100Pwith the plasma membrane either through differential separation of plasma membranes or by biochemical analysis. Cell lysates from trophoblast cell lines and the resulting protein samples from different subcellular locations were collected using differential centrifugation and separated on SDS-PAGE before Western blotting analysis using markers for both the cytoplasmic and membrane components (Figure 3A). Tubulin was used as a cytoplasmic marker [39] and was found to be significantly enriched in the total cell lysate samples compared to the membrane compartment for all samples tested (Figure 3B), highlighting the fact there was little cytoplasmic contamination in the membrane fraction (Figure 3A,B). Caveolin I, a well-established plasma membrane marker [40], on the other hand, was seen greatly enriched in the plasma membrane lane confirming the appropriate isolation of this cell compartment. Figure 3A shows that although the largest proportion of S100P was found in the total cell lysate sample, its presence is clearly seen in the plasma membrane fraction of all cell lines expressing S100P, regardless of the endogenous levels of expression. Quantification of the bands’ intensities in the different fractions for all cell lines tested indicated that there was a significant difference (enrichment was increased by 2.5- to 4-fold) in the levels of S100P in the membrane fraction, when compared to the tubulin marker (Figure 3B). Interestingly, the membranous pool of S100P is characterised by a band which appears to run slower than the proteins from the total cell lysates (Figure 3A) suggesting some type of post-translational changes and an active process to allow the protein to be localized in this cellular compartment. Furthermore, S100P is also present in the fraction of proteins representing the surfaceome of all three trophoblast cell lines (Figure 3C) and primary EVTs (Figure 3E) after biotinylation. However, levels of S100P observed in this fraction are much reduced when compared with those seen in total extracts (Figure 3D) and purification of this pool of proteins, suggesting the presence of S100P in the plasma-membrane/extracellular environment.

### 3.4. Inhibition of Extracellular Cell Surface S100P Results in Significant Reductions in Jeg-3 and HTR8/SVneo Cell Migration and Invasion

Having shown the presence of S100P at the membrane/extracellular cell surface, we next sought to determine if this fraction of S100P contributes to the regulation of trophoblast motility and invasion. This was conducted through a loss of function experiment using an anti-S100P antibody. We and others have previously shown such strategy to inhibit the protein’s extracellular and membrane activities in cancer cells [24,41]. Jeg3 cells and HTR8/SVneo cells were used as they have been shown to be good models for trophoblast migration [42,43].

Jeg-3 cell migration, measured using Boyden chamber assays, was significantly reduced after incubation with the S100P antibody by around 25% (Figure 4A; *p* < 0.001). It is however important to highlight that in keeping with our previous report [9], a greater reduction in motility, around 60% (*p* < 0.001) was achieved by reducing S100P expression via siRNA (Figure 4A) when compared to the mock transfection control. We have previously shown that siRNA treatment can specifically reduce S100P levels in Jeg-3 cells and that this reduction leads to significantly lower levels of cell migration and invasion [9]. These experiments were repeated again here for the purpose of comparison between total loss of function and inhibition of the extracellular pool. This suggests that S100P antibody addition and its action in the extracellular space/membrane can only partially suppress trophoblast migration when compared to a total loss of S100P functions after knock down.

Our previous work has shown that S100P-mediated changes in trophoblast motility are accompanied by changes in the overall actin cytoskeleton and focal adhesion organisation [9]. This is perhaps not surprising since focal adhesion assembly is an important regulator of cellular motility, and the proteins complexed in these structures undergo profound remodelling over time [44]. One such protein, paxillin, is key to the early stages of focal adhesion formation [45]. Therefore, to decipher some of the molecular mechanisms underlying the membrane-associated S100P’s effect on cellular motility, we analysed paxillin and the actin cytoskeleton in both control and anti-S100P antibody-treated Jeg-3 as well as following S100P siRNA treatment (Figure 4C). Untreated or mock treated Jeg-3 cells show a punctate focal adhesion formation/maturation that is characterised by their presence in low numbers and relatively small size (Figure 4C and [9]). Treatment with the anti-S100P antibody had no clear effect on either of these parameters (Figure 4C, Table 1) and did not result in overall changes in the organisation of the actin cytoskeleton (Figure 4C). In contrast, reducing S100P expression via siRNA-mediated knockdown led to significant changes in the architecture of the focal adhesions, resulting in a 77% increase in their number (*p* < 0.0001, Table 1) and equally affected the actin cytoskeleton (Figure 4C) when compared to the mock transfection control, in agreement with what we have previously reported [9].

Similar experiments were carried out in two independent HTR8/SVneo clonal cell lines with high S100P expression (clone 5 and clone 7) as well as a control clone with undetectable levels (clone 3). As previously reported [9], S100P expression in both of the HTR8/SVneo cells (clone 5 and clone 7) leads to a significant 2–3-fold increases in motility when compared to the control counterpart (*p* < 0.0001, Figure 5A). Treatment of control HTR8/SVneo clone 3 cells with the anti-S100P antibody did not affect their motility (*p* = 0.62; Figure 5A), validating the antibody’s specificity., However, migration of either HTR8/SVneo clone 5 or clone 7 cells was significantly reduced by around 35% (*p* < 0.0001; Figure 5A) albeit not to the level observed in clone 3 cells.

Analysis of focal adhesion organisation and numbers in the HTR8/SVneo trophoblast cell clones revealed that whilst clone 3 cells demonstrated a large number of paxillin-containing focal adhesion clusters, high expression of S100P in the same cell background (clone 5 and clone 7) resulted in more diffuse or punctate focal adhesion formation/maturation, further characterised by their presence in small numbers (*p* = 0.0078 and *p* = 0.0069, Table 1) and relatively small size (Figure 5C). The anti-S100P antibody did not affect the focal adhesions or the overall architecture of the actin cytoskeleton in either of the S100P-expressing HTR8/SVneo clones (clones 5 and 7, Figure 5C, Table 1) or the non-expressing control counterpart.

Given the well-established role for S100P in promoting cell invasion in both cancer cells [16,23] and trophoblast cells [9], we next wanted to assess whether the extracellular/membrane bound S100P pool of proteins could regulate the invasiveness of Jeg-3 (Figure 4B) and HTR8/SVneo cells (Figure 5B). Similar experiments were carried out using cells starved for serum for 24 h prior to seeding with or without S100P antibody on congealed matrigel in Boyden chambers and allowed to invade the gel in the membrane.

Jeg-3 cell controls as well as the mock-treated counterparts demonstrated good abilities to invade through matrigel (Figure 4B; *p* = 0.989). S100P antibody treated Jeg-3 cells demonstrated a significant diminution in their ability to invade into the matrigel (~45% *p* = 0.0002; Figure 4B). This reduction was, however, smaller than that observed when S100P levels were specifically and significantly reduced by siRNA4, as such knock down resulted in a 75% reduction in their ability to invade (*p* < 0.0001; Figure 4B and [9]). This again indicates that despite the ability of the S100P antibody to lower the invasive traits of Jeg-3 cells, such treatment did not fully abolish the S100P-dependent effects.

As shown previously, expression of S100P in the HTR8/SVneo cell background resulted in a dramatic increase in cell invasion [9], leading to a 4- to 6-fold increase in the ability of clone 5 and clone 7 cells to invade, respectively when compared to the control clone 3 counterpart (*p* < 0.0001, Figure 5B). Whilst mock treatment did not have any effects on the cells ability to reach the outer side of the chamber [9], addition of S100P antibody to clone 5 or clone 7 HTR8/SVneo trophoblasts expressing S100P resulted in a significant 35% reduction in their invasive traits (Figure 5B; *p* < 0.0001). On the other hand, addition of S100P antibody to the non-expressing S100P clone 3 cells did not result in any changes in invasion (Figure 5B; *p* = 0.99). Interestingly and similarly to the migration work, treatment with the S100P antibody did not result in the complete abolishment of the S100P-specific invasive properties as numbers of both HTR8/SVneo clone 5 and clone 7 cells allowed to invade in the presence of the S100P antibody were still 3-fold higher than that of the control clone 3 cells (Figure 5B; *p* < 0.0001).

All together, this data demonstrates that inhibiting the activity/presence of extracellular/membrane bound S100P in cultured Jeg-3 and HTR8/SVneo trophoblastic cells clones 5 and 7 causes a decrease in their ability to both migrate and invade, but that such inhibition is not fully reverted through a mechanism which is focal adhesion independent, indicating that other processes are still at play to promote cell motility/invasion.

### 3.5. Inhibition of Extracellular Cell Surface S100P Reduces Motility and Invasion of Primary EVTs

Having established the importance of extracellular/membrane bound S100P in the motility and invasion of the different trophoblast cell lines tested in this work so far, we next sought to further demonstrate whether similar observations could be made when using human primary first trimester EVT trophoblasts (Figure 6).

Primary EVT-like trophoblasts were isolated as described in methods, seeded onto fibronectin coated plates, and grown for 24–48 h. Immunofluorescence staining of isolated cells using the HLA-G marker demonstrated that more than 80–90% of the cells were differentiated EVT like cells [8].

Motility and invasion experiments were carried out with or without the presence of the S100P antibody, prior to seeding primary EVT onto untreated Boyden chambers. Anti-S100P antibody treatment of primary EVT led to a reduction in both migration (~35%, *p* < 0.0001; Figure 6A) and invasion (~40%, *p* < 0.0001; Figure 6B) compared to the mock treated samples.

We also investigated whether the anti-S100P antibody affected the overall organisation of focal adhesions and the actin cytoskeleton (Figure 6C). Primary EVTs presented a significant display of focal adhesions around the cell cortex, as demonstrated by the large collection of paxillin clusters at the cell periphery. These elongated complexes were found to be capping the end of cortical actin filaments in all conditions tested. Upon treatment with S100P antibody, there were no significant changes in the number of focal adhesions (Figure 6C), after quantitative analysis (Table 1; *p* = 0.9249), and no clear changes in actin cytoskeletal organisation observed (Figure 6C).

All together the data demonstrates that inhibiting the activity/presence of extracellular/membrane bound S100P in all trophoblast cells tested, whether lines such as Jeg-3 and HTR8/SVneo cells or primary EVT cells, leads to significant reductions in cellular motility and invasion. These reductions were however not fully reverted to migration/invasion seen when there are no detectable levels of S100P expression (when using HTR8/SVneo cells) or when total S100P levels were more than 80–90% ablated (in Jeg-3 cells after siRNA delivery [9]). Our data suggests that different pools of S100P may contribute to the regulation of cellular migration and invasion processes with district mechanisms, one which includes the overall organisation of either the actin cytoskeleton or the focal adhesions but is not inhibited by S100P antibody addition.

### 3.6. Inhibition of Extracellular Cell Surface S100P Does Not Affect Jeg-3 and HTR8/SVneo Cell Viability and Proliferation

High levels of S100P have been linked to increased cell viability and cellular proliferation in JAR cells, but not other choriocarcinoma cell lines (Jeg-3 or BeWo) [26] or in extravillous trophoblast (HTR8/SVneo) [9]. We therefore explored whether the anti-S100P antibody’s effects on motility and invasion could be explained by changes in cell viability. As shown in Figure 7, S100P antibody-treated Jeg-3 (Figure 7A) and the different HTR8/SVneo clonal cells (S100P expressing clones 5 and 7 as well as the control counterpart clone 3, Figure 7B) resulted in similar cell numbers to mock treated and untreated counterparts at both 24 h and 48 h, indicating that neither cell viability nor proliferation were compromised by the presence of the antibody.

## 4. Discussion

It is shown here that pools of S100P were found localised in the proximity of the plasma membrane following immunohistochemistry staining of first trimester human placental sections (Figure 1) as well as through immunofluorescence of different trophoblast cell lines (Figure 2A,B) and primary EVTs (Figure 2C). This observation was evidenced by the clear localisation of some of the S100P pools in close proximity to where cell surface/membrane HLA-G [46,47,48] could be found. Immunofluorescence work using either the membrane marker Nile red [34,35,36] or the receptor integrin β1 [5,49] also provides strong evidence suggesting that some pools of S100P were localised near the plasma membrane. The use of the plot profiling and ROI technique is well characterised as an efficient method to evidence the colocalisation of specific complexes for both immunofluorescence and immunohistochemistry staining as well as for membrane localisation [50,51,52]. The presence of S100P close to the plasma membrane has never been reported before in the context of trophoblast cells, either in vivo or in different cell lines and primary EVT cultures. However, our group and others have recently demonstrated the presence of a S100P in the plasma membrane of breast cancer cells [23,24] and in epithelial cells of the suprabasal and upper layers of the human oesophageal epithelium by immunohistochemistry [53]. How S100P could be localised in this part of the cell is currently not known. Interactions of this protein with either IQGAP1 or ezrin have been reported [21,54]. Given that either of these molecules are found localised close to the plasma membrane [50,55] albeit in the intracellular cortical side where they are known to regulate the overall architecture of the actin cytoskeleton, it is plausible to hypothesise that some of the S100P pools observed in this region of the cells may actually be intracellular fractions associated with these proteins.

The concept of S100 proteins being found at the plasma membrane is however not necessarily surprising as other S100 proteins have been shown to bind specific membrane-associated partners. For instance, S100A4 as well as S100A10 have been reported to interact with annexin 2 [56,57]. However, no interaction between S100P and annexin A2 has yet been found [23]. Other S100 proteins such as S100A1, S100B, S100A6, and S100A11 have also been shown to play important roles in membrane repair and exocytosis via interactions with members of the phospholipid membrane annexin family [58,59,60,61]. In trophoblast cells, the association of Annexin A2 with the S100A10 complex has recently been shown in placental membrane [62] providing further support for the presence of S100 proteins in this cellular environment. It is currently unknown whether S100P would be able to associate with any of these different factors to facilitate its localisation and effects at the plasma membrane.

Signals for S100P were detected in cell membrane preparations of all cell lines by cell fractionation (Figure 3A) and by biotinylation purification for HTR8/SVneo /Jeg-3 cell lines as well as the primary EVT. Both of these techniques have been extensively used to characterise the presence of numerous factors within the membrane/extracellular cell surface in trophoblast cells [63,64,65] and others [66]. Biotinylation allows for the pull down of factors that are either associated with the cell surface or the extracellular matrix. Thus, it is possible that some of the S100P detected may be present in the extracellular matrix since it has been shown to be secreted in this space via both autocrine and paracrine processes, at least in cancer cells [18,67]. However, there are no clear molecular pathways to facilitate this secretion since S100P lacks a conventional secretory signal sequence. There is currently no evidence that S100P can indeed be secreted from trophoblasts and is not the result of cell leakage, but this data suggests that some of the pools of S100P detected are found in the extracellular environment.

Purification of the plasma membrane also demonstrates some of the S100P pools are directly associated with such structures although it is not clear at this stage how this would be achieved. S100P has been shown to interact with different extracellular cellular receptors. Among others, extracellular S100P is thought to bind to the receptor for advanced glycation end products (RAGE) with affinity between S100P and the domain V receptor lying in the micromolar range [62,67,68,69]. S100P has also been reported to interact with IL-11 [70] whereby both bind to one another as well as the IL-11 receptor and gp130 in the formation of a quaternary complex. The identification of S100P as part of the membrane fraction isolated after sucrose separation indicates that the association of the protein with this domain is stable and can withstand cell rupture and centrifugation. It is therefore unlikely that the pools of S100P detected could be the result of the transient ligand receptor interactions listed here and suggests a more permanent binding partner. The changes in migration pattern for S100P after SDS-PAGE separation in the membrane fraction also suggest that some post-translational changes may be taking place which would facilitate its integration within lipid moiety. Whilst it is beyond the scope of this work to determine the exact process, our preliminary work suggests that specific resides on S100P are likely candidates for lipidation. In this context, our current predictions indicate that different residues within S100P are potential substrate for N-myristoylation and S-farnesylation. Interestingly, the residues in questions would likely be located at the *C*-terminal part of the protein, a region we have recently shown to be important for cancer cell motility and invasion [23,24]. In all of these accounts, the presence of S100 proteins at the plasma membrane were primarily characterised in cancer cells but the presence of S100P in the plasma membrane of non-carcinogenic cells is yet unreported. We provide new evidence that S100P can be found associated with the external region of the plasma membrane in all trophoblast cell lines tested, as well as in primary EVTs.

The main finding in the present work suggests the identification of an unexplored pathway by which a specific pool of S100P associated with the plasma membrane and extracellular space promotes both cell invasion and cell migration, since this pathway can be inhibited by a S100P-specific antibody added to the medium. It is unlikely that this treatment affects intracellular S100P given that antibodies do not cross cell membranes [71]. The presence of extracellular S100P in promoting cell motility and invasion has not been reported in the context of trophoblast cells and has only recently been published in relation to breast cancer cell migration and invasion [23,24]. Others have demonstrated that exogenous S100P has also been shown to increase the secretion of matrix metalloproteinase 9 (MMP-9) in pancreatic cancer cells although a role in cellular invasion was not shown [41].

How the presence S100P in the extracellular/membrane bound fraction is responsible for promoting these effects is currently not known but appears not to plays a role in focal adhesion changes that are seen when high levels of S100P are expressed in trophoblast cells (Figure 4 and Figure 6) or in cancer cells [16]. Interestingly, whilst S100P antibody addition was successful in inhibiting some but not all of the motility and invasion properties, the reduction reported was not complete. Cells treated with S100P antibody were still more efficient in migrating and invading than those where levels of S100P were reduced (in Jeg-3 after siRNA delivery (Figure 4; Ref. [9]) or when expression was not detectable in HTR8/SVneo cells clone 3 (Figure 5). This data suggests that S100P can promote motility and invasion through at least two independent pathways. Staining of the actin cytoskeleton and quantification of paxillin-containing focal adhesions supported this hypothesis, as inhibition of S100P by antibody addition was found not to significantly affect the overall actin structure nor the number of focal adhesion clusters. This is in distinct contrast to those seen following total reduction in S100P in cells where clear changes could be seen (Figure 4 and Figure 5) and as we have reported previously [9]. The mechanisms linking S100P levels to changes in overall cytoskeletal and focal adhesion structures have not been explained Molecules such as ezrin [8,54], IQGAP1 [21] or non-muscle myosin IIA [16] have all been shown to act as S100P binding partners. Their roles in regulating cellular motility and invasion thought regulation of the actin cytoskeleton and focal adhesions, mainly in cancer cells are well established [72,73,74].

Interestingly, whilst ezrin and IQGAP1 have been found in the vicinity of the plasma membrane, they remain intracellular but have been shown to regulate the externalisation of different interacting partners [75,76] and could therefore offer another potential pathway for the presence of S100P in this location. Whilst inhibiting S100P at the extracellular/membrane-bound compartment was shown not to affect focal adhesions and the overall actin organisation, it is important to note that recent studies in relation to externalised S100 proteins affecting the cytoskeleton have been reported. For instance, S100A10 was shown to be required in HeLa cells for the overall organization of actin stress fibres and the formation of focal adhesions via the Rac1 pathway [77] and S100A11 is found to be enriched and responsible for regulation of actin rich pseudopodia in metastatic cells [78] as well as regulating the rate of actin polymerization when associated with Annexin A2 [79]. Given that Annexin A2 is readily expressed in trophoblast cells and promotes their migration [80], it is possible to speculate that in the case of our experiments, S100P antibodies may act as an agonist against the membrane-bound pool and prevent the activation of the membrane receptors annexin 2, as has been shown for other S100 proteins [81]. We are, however, aware that no interaction between S100P and annexin A2 has yet been found [23] in a cancer setting.

These findings shed new light on at least two newly characterized pathways by which S100P promotes trophoblast cellular motility and invasion. One where cellular S100P levels involve the remodelling of focal adhesion and another, an extracellular pathway which is focal adhesion independent. Both pathways may lead to the identification of novel targets that may explain why significant numbers of confirmed human pregnancies suffer complications through poor placental implantation.

## Figures and Tables

**Figure 1 biomolecules-13-01231-f001:**
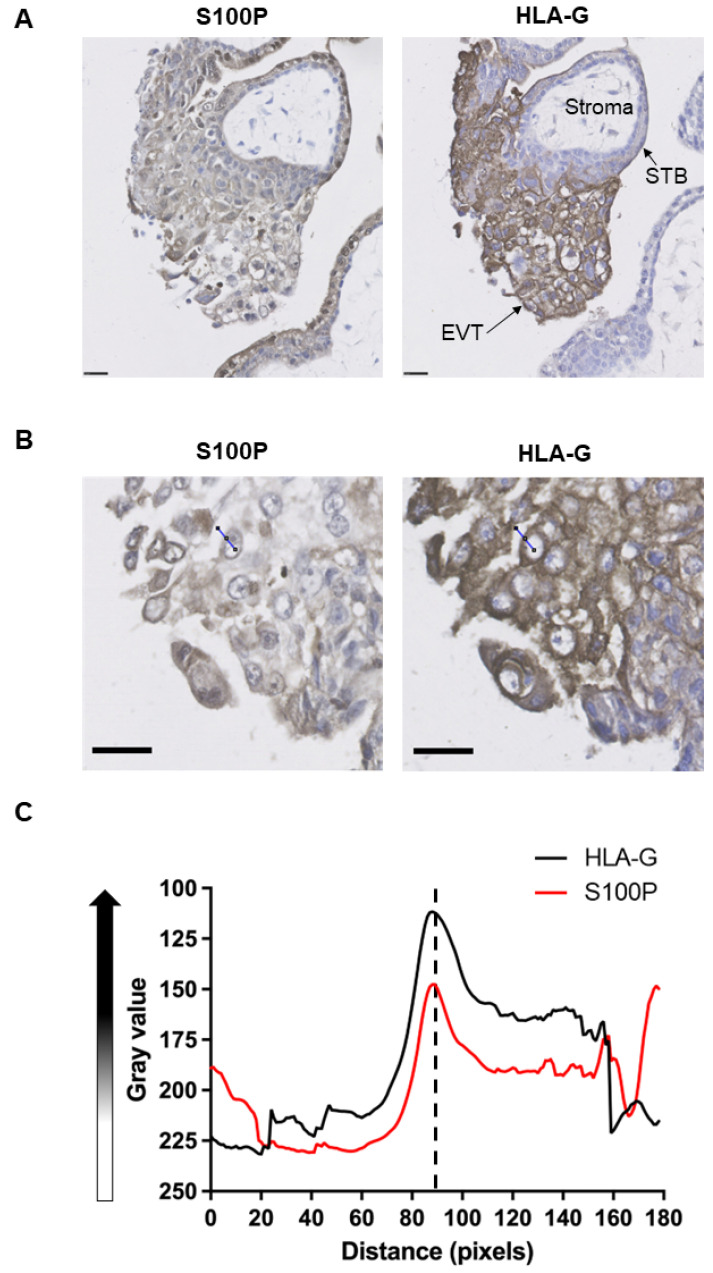
Pools of S100P are localised close to the plasma membrane in trophoblasts including extravillous trophoblasts in placental first trimester tissues. Serial sections of first trimester human placental tissues were stained using either a rabbit monoclonal S100P antibody or mouse monoclonal HLA-G antibody as described in Methods (**A**,**B**). Arrows indicate extravillous trophoblasts (EVT) and syncytiotrophoblasts (STB). Stroma is also highlighted. Bar corresponds to 150 μm (Black bar). Immunohistochemistry staining of matched S100P and HLA-G tissues (**B**) were quantified by drawing a region of interest (ROI) centred on cell membranes using ImageJ. (**C**) Plot of the length of the ROI in pixels against the mean grey value recorded for the ROI. Lower grey values indicate higher concentrations of the protein of interest. Graph is a summation of 30 different regions from multiple first trimester placental samples. Midpoint of each graph depicts the assumed plasma membrane.

**Figure 2 biomolecules-13-01231-f002:**
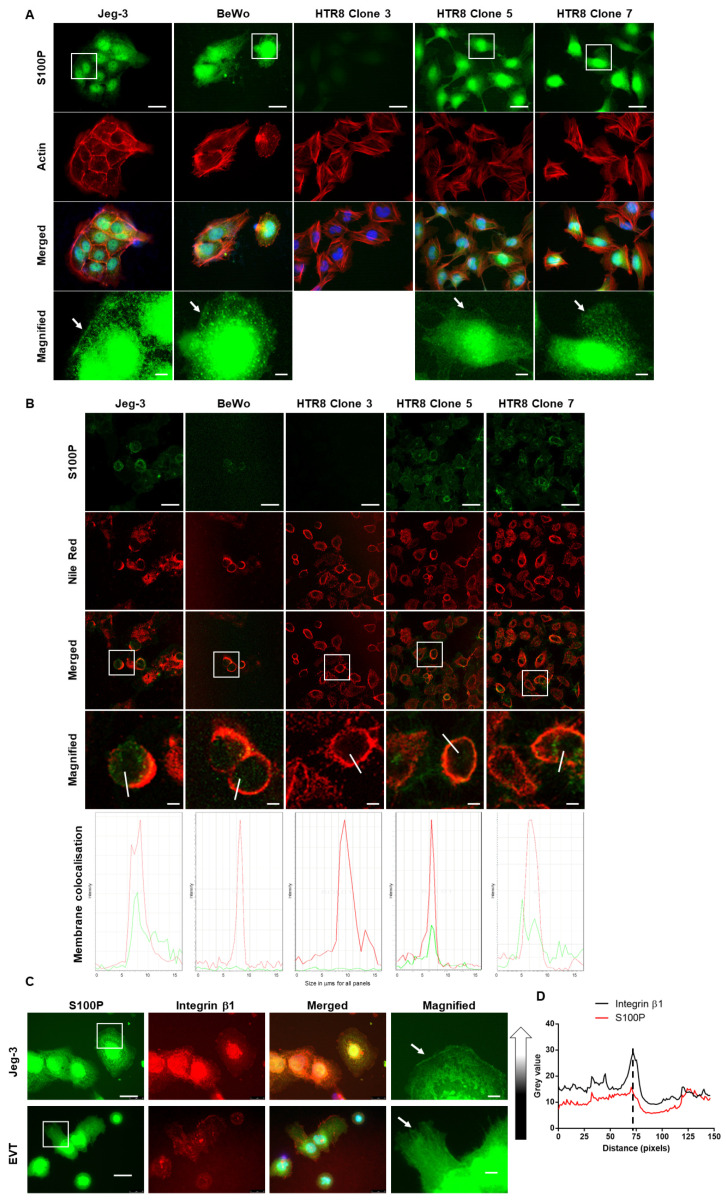
Pools of S100P are localised near to the plasma membrane in all trophoblast lines and primary EVTs cells. HTR8/SVneo S100P expressing clone 5 and clone 7 cells, as well as the respective negative control (clone 3) (**A**,**B**), Bewo cells (**A**,**B**) Jeg-3 cells (**A**–**C**), as well as primary EVT cells (**C**) were seeded on fibronectin-coated coverslips and grown for 48 h. Cells were fixed, permeabilized and stained for S100P and actin (**A**) or S100P and integrin β1 (**C**). In parallel, cells were incubated with S100P antibody and Nile red prior to fixation and further processing for staining (**B**). All cells were mounted and viewed using epifluorescence (**A**,**C**) or confocal microscopy (**B**). Images in the last column correspond to the enlarged regions of the highlighted cells (White box). White arrows correspond to specific regions of cells refered to in the data interpretation. Bar corresponds to 25 μm in the wide views and 5 μm in the zoomed in regions (White bar). Colocalisation immunofluorescence staining of S100P and Nile red (**B**) or S100P and integrin β1 (**C**) in cells were quantified by drawing a region of interest (ROI shown as a white bar across cells) through cell membrane regions using Leica Application Suite X or ImageJ, respectively. Quantification of fluorescent profiles in lower panel (**B**) correspond to a single analysis of a representative cell. (**D**) Plot of the length of the ROI in pixels against the mean grey value recorded for the ROI. Higher grey values indicate white pixels and therefore higher concentration of the proteins of interest, whereas lower grey values indicate black pixels. Graph is a summation of 30 different regions from 3 independent experiments. Midpoint of each graph depicts the assumed plasma membrane.

**Figure 3 biomolecules-13-01231-f003:**
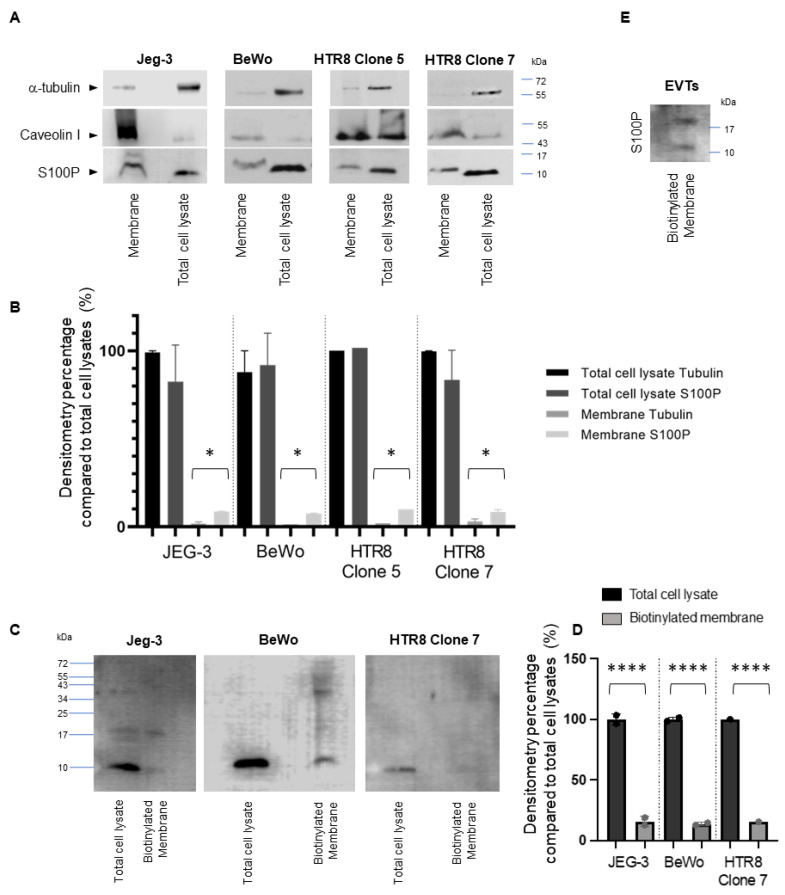
Characterisation of membrane/cell surface S100P in all trophoblast lines. Plasma membrane fractions and total cell lysates from different trophoblastic cell lines (Jeg-3, BeWo and HTR8/SVneo S100P expressing Clone 5 and Clone 7) were isolated and samples solubilised in Laemmeli buffer. Equal loading for the respective fractions of proteins were separated by SDS-PAGE electrophoresis. Western blotting was carried out and membranes probed for S100P, caveolin I and α-tubulin and cropped images are presented (**A**). Quantification using densitometry analysis is presented to correspond to the percentage of signal obtained from the membrane compared to the proportion seen in the total cell extract (for tubulin or S100P, **B**). Cell surface S100P was isolated using biotinylation of live cultures of trophoblast cell lines (**C**) and primary EVTs (**E**). Pulldown fractions were analysed by Western blotting for S100P (**C**,**E**). Quantification using densitometry analysis is presented to correspond to the percentage of S100P signal obtained from the extracellular environment compared to the proportion seen in the total cell extract, (**D**). Data are presented as percentage seen in the plasma membrane (mean ± SEM of 3 independent experiments). * *p* < 0.05 or **** *p* < 0.0001 compared to total cell lysates (one way-ANOVA).

**Figure 4 biomolecules-13-01231-f004:**
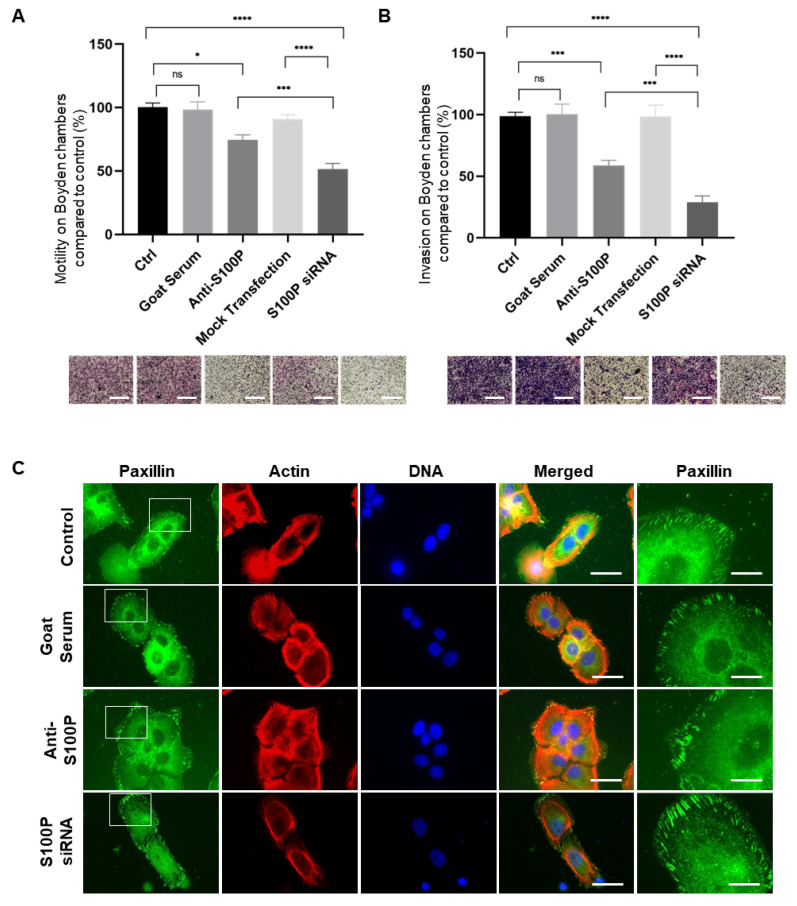
Inhibiting extracellular/membrane bound S100P impairs motility and invasion of Jeg-3 trophoblast cells without changes in focal adhesion clustering. Jeg-3 cells were treated with S100P antibody, S100P siRNA or their respective mock-controls prior to seeding into Boyden chambers without matrigel for motility (**A**) or with matrigel for invasion (**B**) and allowed to migrate for 24 h prior to fixation and staining using the May Grunwald-Giemsa kit for labelling of both nuclei and cytoplasm. Five random fields were quantified for each chamber. Data are presented as percentage (mean ± SEM of 4 independent experiments) migration/invasion relative to their respective controls. * *p* < 0.05, *** *p* < 0.001, **** *p* < 0.0001 and ns (not significant, *p* > 0.05) compared to control and mock treated (one way-ANOVA). Images of representative fields of motility/invasion assays were taken with the EVOS XL Cell Imaging System at ×20 magnification. Bar corresponds to 100 μm. Jeg-3 cells were seeded on fibronectin-coated coverslips prior to treatment with S100P antibody, S100P siRNA, or mock-control for a further 24 h before fixing and staining for the focal adhesion marker paxillin and the cytoskeletal marker actin (**C**). Cells were mounted and viewed using epifluorescence microscopy. Images on the last column correspond to the enlarged regions of the highlighted cells (White box). Bar corresponds to 50 μm in the wide views and 10 μm in the zoomed in regions (White bar).

**Figure 5 biomolecules-13-01231-f005:**
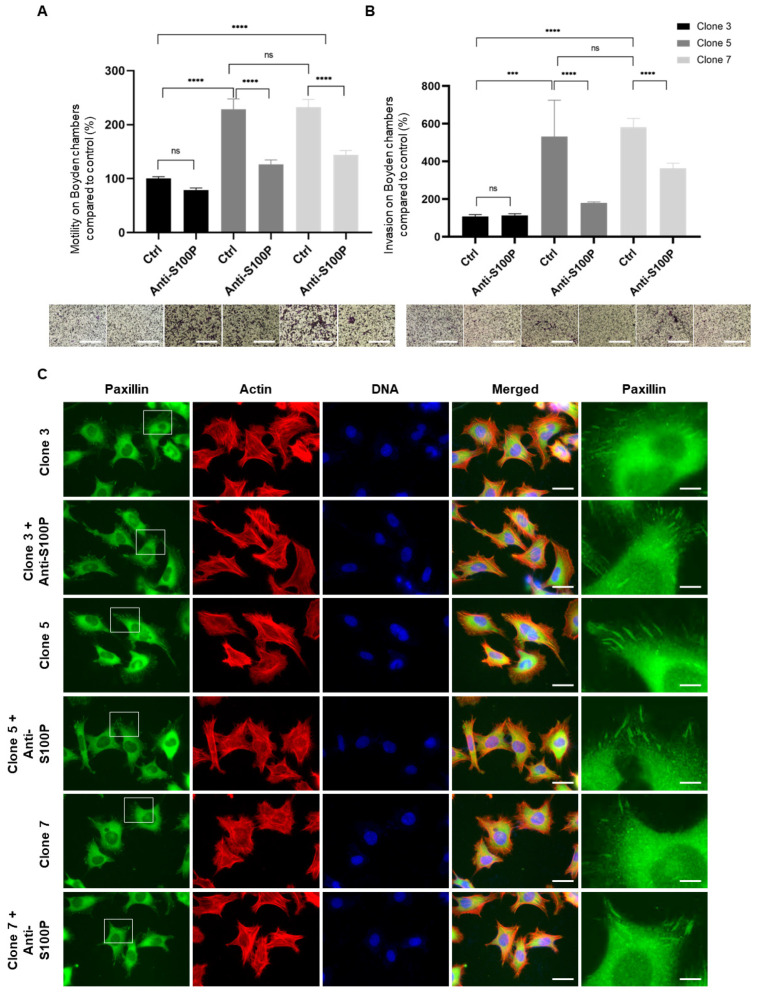
Specific inhibition of extracellular/membrane bound S100P impairs motility and invasion of HTR8/SVneo S100P expressing cells but without changes to focal adhesion clustering. Two independent HTR8/SVneo S100P-expressing cells (clone 5 and clone 7) and their control counterparts (clone 3) were treated with S100P antibody or mock-control prior to seeding into Boyden chambers without matrigel for motility (**A**) or with matrigel for invasion (**B**) and allowed to migrate for 24 h prior to fixation and staining using the May Grünwald-Giemsa kit for labelling of both nuclei and cytoplasm. Five random fields were quantified for each chamber. Data are presented as means ± SEM of 4 independent experiments relative to controls (percentage) from 4 replicate wells for each set of conditions. *** *p* < 0.001 and **** *p* < 0.0001 and ns (not significant) *p* > 0.05 compared to control and mock treated (one way-ANOVA). Images of representative fields of motility/invasion assays were taken with the EVOS XL Cell Imaging System at ×20 magnification. Bar corresponds to 100 μm (White bar). Two independent HTR8/SVneo S100P-expressing clone 5 and clone 7 trophoblast cells and the non-expressing counterpart clone 3 were seeded on fibronectin-coated coverslips prior to treatment with S100P antibody or mock-control for a further 24 h prior to fixation, permeabilisation, and staining for the focal adhesion marker paxillin and the cytoskeletal marker actin (**C**). Cells were mounted and viewed using epifluorescence microscopy. Images on the last column correspond to the enlarged regions of the highlighted cells (White box). Bar corresponds to 50 μm in the wide views and 10 μm in the zoomed in regions (White bar).

**Figure 6 biomolecules-13-01231-f006:**
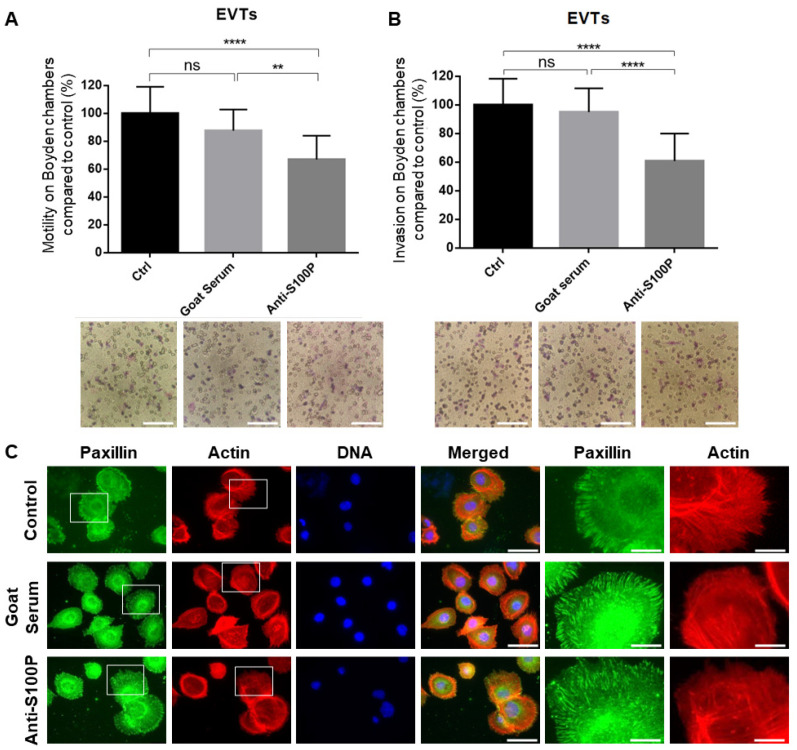
Specific inhibition of extracellular/membrane-bound S100P impairs motility and invasion of human primary EVT cells without changing focal adhesion clustering. Primary human EVTs cells were extracted from first trimester placenta samples and allowed to recover for 24 h. Cells were treated with S100P antibody or mock-control (goat serum) prior to seeding into Boyden chambers without matrigel for motility (**A**) or previously coated with matrigel for invasion (**B**) and allowed to migrate for 16 h prior to fixation and staining using the May Grϋnwald—Giemsa kit for labelling of both nuclei and cytoplasm. Five random fields were quantified for each chamber. Data are presented as means ± SEM of 4 independent experiments relative to controls (percentage) from 4 replicate wells for each set of conditions. ** *p* < 0.01 and **** *p* < 0.0001 and ns (not significant) *p* > 0.05 compared to control and mock treated (one way-ANOVA). Images of representative fields of motility/invasion assays were taken with the EVOS XL Cell Imaging System at ×20 magnification. Bar corresponds to 100 μm (White bar). Primary human EVTs cells were seeded on fibronectin-coated coverslips prior to treatment with S100P antibody or mock-control for a further 24 h prior to fixation, permeabilisation, and staining for the focal adhesion marker paxillin and the cytoskeletal marker actin (**C**). Cells were mounted and viewed using epifluorescence microscopy. Images on the last two rows correspond to the enlarged regions of the highlighted cells (White box). Bar corresponds to 50 μm in the wide views and 10 μm in the zoomed in regions (White bar).

**Figure 7 biomolecules-13-01231-f007:**
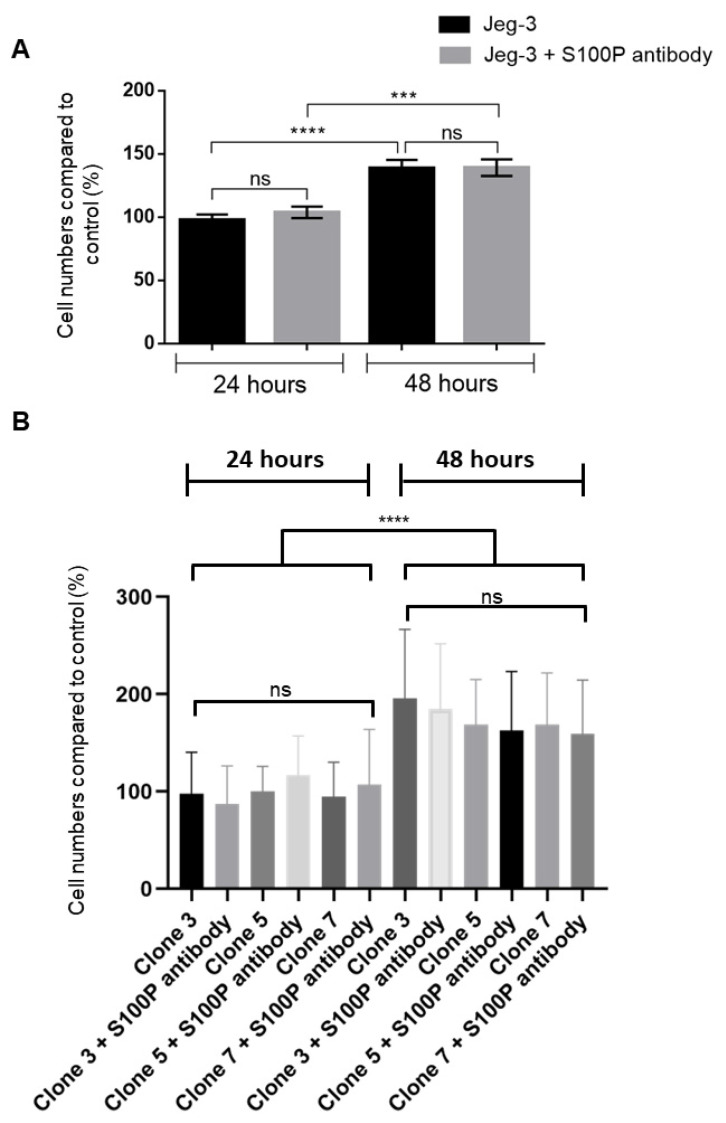
Inhibiting extracellular/membrane-bound S100P does not affect Jeg-3 and HTR8/SVneo trophoblast cell proliferation. Jeg-3 cells (**A**) and high S100P expressing HTR8/SVneo clone 5 and clone 7 and their counterpart control clone 3 cells (**B**) were seeded into 24 well plates and treated with S100P antibody or mock-control for 24 and 48 h before assessing cell viability using trypan blue exclusion. Data are presented as percentage of cells (mean ± SD of 3 independent experiments relative to 24 h untreated controls from 3 replicate wells for each set of conditions (one way-ANOVA). *** *p* < 0.001, **** *p* < 0.0001, and ns (not significant, *p* > 0.05).

**Table 1 biomolecules-13-01231-t001:** Inhibiting extracellular S100P in trophoblast cells by addition of S100P antibody does not lead to significant changes in the number of focal adhesions per cell but regulating overall S100P cellular levels do.

Cell Lines	Percentage Focal Adhesions per Cell ± SEM (n = 40)	*p* Value ^a^	*p* Value ^b^
Jeg-3 control	100 ± 11.17		
Jeg-3 treated with S100P antibody	101.83 ± 7.93	0.9861	
Jeg-3 treated with S100P siRNA	177.08 ± 4.54	*p* < 0.0001	*p* < 0.0001
HTR8/SVneo clone 3 control	100 ± 3.412		
HTR8/SVneo clone 3 treated with S100P antibody	101.18 ± 2.93	0.9988	
HTR8/SVneo clone 5 control	72.67 ± 7.25		*p* < 0.01
HTR8/SVneo clone 5 treated with S100P antibody	80.12 ± 5.93	0.9120	
HTR8/SVneo clone 7 control	75.11 ± 4.10		*p* < 0.01
HTR8/SVneo clone 7 treated with S100P antibody	76.51 ± 3.53	0.9981	
Primary EVTs control	100 ± 6.21		
Primary EVTs treated with S100P antibody	103.08 ± 7.06	0.9249	

Jeg-3 cells, HTR8/SVneo clones and primary EVTs, as well as cells mock-treated or with addition of S100P antibody for 24 h were fixed and stained for paxillin and actin after seeding on fibronectin-coated coverslips. Data shown are means ± SEM corresponding to the average number of focal adhesion-containing paxillin observed per cell, presented as percentage of control. Untreated and mock treated controls were found to not be statistically significant (*p* > 0.05) [9]. ^a^
*p*-value obtained from one-way ANOVA where total number of focal adhesions present in Jeg-3, HTR8/SVneo clones or primary EVTs control cells were compared to S100P antibody/s100P siRNA-treated counterparts. ^b^
*p*-value obtained from one-way ANOVA where total number of focal adhesions present in Jeg-3 control cells were compared to S100P siRNA-treated counterparts, or HTR8/SVneo clone 3 compared to either HTR8/SVneo clone 5 or HTR8/SVneo clone 7 counterparts.

## Data Availability

The data presented in this study are available in this article and Appendix A.

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
