# Peer review of "An Extracellular/Membrane-Bound S100P Pool Regulates Motility and Invasion of Human Extravillous Trophoblast Lines and Primary Cells"

_biomolecules, 2023, doi:10.3390/biom13081231_

Round 1
Reviewer 1 Report
Lancaster et al. describe the characterization of S100P in the plasma membrane of placental cells using a combination of cell lines, primary tissue, and biochemical approaches. While the methodology seems appropriate to answer some of the questions posed by the authors, the quality of the results and some of their interpretations make the manuscript difficult to follow and the conclusions difficult to accept. Due to these reasons, this reviewer cannot support the publication of the manuscript. Below are some examples of the main issues, in no particular order:
1) Why does a cytosolic protein without a signal peptide or transmembrane domain appear in the extracellular compartment? Is there any strong evidence? Even in the discussion (L682-685), authors discuss the possibility that S100P binds to partners at the plasma membrane's cytosolic face. Then how do they explain the experiments with anti-S100P exogenously added without affecting focal adhesion or actin remodeling? In the end of the manuscript, these issues are discussed, but there is no solid explanation or mechanistic experiments addressing them. The lack of reliable data makes the initial hypothesis very weak, when compared with other simpler alternatives (intracellular role, for example).
2) After performing immunofluorescence in the different cells with triton X, the authors conclude from the images that S100P localizes at the plasma membrane using regular epifluorescence. Tests for plasma membrane localization should not be conducted in this manner. The staining should take place in unpermeabilized cells, and it is best to use confocal microscopy for this.
3) There is a discrepancy between the conclusions drawn from Figure 1 analysis and what is observed in the figures. Despite line analysis, the IHC staining does not provide enough resolution to confidently demonstrate that the signal originates from the plasma membrane.
4) There are no conclusive results from the cell fractionation studies. S100P blot differences between total lysate and membranes in some samples (BeWo and HTR8) are very similar to those shown for tubulin, suggesting that maybe the signal for S100P is still coming from the cytosol, not entirely from the membrane fraction. It is also unfortunate that the biotinylation assays lack quality and positive and/or negative controls to be confidently conclusive. The authors do not explain why S100P runs differently in membrane fractions or why different bands are observed.
5) Based on the Giemsa staining pictures, it's extremely difficult to believe that the S100P siRNA condition has significantly less signal than the anti-S100P condition.
6) What is the origin of clones 3 and 7 of HTR8 cells?
None
Author Response
See attached document

Reviewer 2 Report
In this paper, Lancaster and colleagues analyze the expression and function of S100P proteins in trophoblast cells and cell lines. Their main conclusion is that S100P proteins are expressed in trophoblast cells, and at least a fraction of them is associated to the plasma membrane, apparently on the outside. Inhibition of S100P proteins by antibody blockage or siRNA decreases cell motility and invasion without compromising the architecture of the adhesions or their ability to proliferate.
The topic of this paper is interesting, but the findings do not entirely support the conclusions of the study and some crucial controls and experiments are missing.
MAJOR POINTS
- In this time and era of research on S100P, there is no real reason to lump them all into a large group. The authors need to provide some specifics as to which S100 proteins are the most crucial ones in this cellular model is required.
- In general, images throughout the paper are of low quality and lack crucial controls. For example, S100 localization in Fig. 2 is very hard to discern. Given that the authors have S100 siRNA available (Fig. 4), they should at least provide evidence that the signal of the antibody used in Fig. 2 goes significantly low in siRNA-treated cells.
- Fig. 3B is very unconvincing. With the exception of the BeWo blot (middle one) the biotinylated band at 10 kD is barely observable. This needs further work and quantification.
- In Fig. 4A/B, scrambled siRNA controls are required.
- Fig. 2A, 4C, 5C, 6C, actin staining looks really bad and saturated, which is not acceptable particularly in light of the much better quality of the paxillin images next to some of these.
- Experiments with just one clone of S100-expressing cells are not acceptable. Pools of at least 5 clones are required.
- Conceptually, what is the meaning of S100P antibodies blocking migration? There are a number of very interesting possibilities, and the authors need to at least initiate one of the following: i) that antibodies are agonistic and membrane-bound S100 proteins are actively signaling, even if it is through clustering of other membrane receptors; ii) that the antibodies block S100P from functioning as receptor/ligand for other membrane-bound proteins in the same or neighbor cells.
- Are there S100P proteins in extracellular vesicles?
- Western blots throughout the paper need to be quantified.
Author Response
See attached document

Round 2
Reviewer 1 Report
I would like to thank the authors for their efforts trying to improve the manuscript. Unfortunately, I do not think that the new experiments address my concerns. The lack of quality of the figures (which makes interpretation very subjective), legends that are very difficult to follow and the choice of the approaches used to answer the main questions of the manuscript still raise the same issues highlighted in the first round of revisions.
English needs to be revised.
Reviewer 2 Report
The authors have satisfied most of my concerns.
English is largely okey.